# (−)-Adaline from the *Adalia* Genus of Ladybirds Is a Potent Antagonist of Insect and Specific Mammalian Nicotinic Acetylcholine Receptors

**DOI:** 10.3390/molecules27207074

**Published:** 2022-10-20

**Authors:** David P. Richards, Rohit N. Patel, Ian R. Duce, Bhupinder P. S. Khambay, Michael A. Birkett, John A. Pickett, Ian R. Mellor

**Affiliations:** 1School of Life Sciences, University of Nottingham, University Park, Nottingham NG7 2RD, UK; 2Protecting Crops and the Environment, Rothamsted Research, West Common, Harpenden AL5 2JQ, UK; 3School of Chemistry, Cardiff University, Cardiff CF10 3AT, UK

**Keywords:** nicotinic acetylcholine receptor, (−)-adaline, *Adalia bipunctata*, *Adalia decempunctata*, patch clamp, voltage clamp

## Abstract

Ladybird beetles (Coleoptera: Coccinellidae) possess strong chemical defences that are secreted in response to stress and are also found on the coating of eggs, which are rich in alkaloids that are responsible for their toxicity to other species. Recent studies have shown that alkaloids from several species of ladybird beetle can target nicotinic acetylcholine receptors (nAChRs) acting as receptor antagonists. Here, we have explored the actions of (−)-adaline, found in the 2-spot (*Adalia bipunctata*) and 10-spot (*Adalia decempunctata*) ladybirds, on both mammalian (α1β1γδ, α7, α4β2, α3β4) and insect nAChRs using patch-clamp of TE671 cells and locust brain neurons natively expressing nAChRs, as well as two-electrode voltage clamp of *Xenopus laevis* oocytes recombinantly expressing nAChRs. All nAChR subtypes were antagonised by (−)-adaline in a time-dependent, voltage-dependent and non-competitive manner with the lowest IC_50_s at rat α3β4 (0.10 μM) and locust neuron (1.28 μM) nAChRs, at a holding potential of −75 mV. The data imply that (−)-adaline acts as an open channel blocker of nAChRs.

## 1. Introduction

Ladybirds (Coleoptera: Coccinellidae) comprise a family of brightly coloured and patterned beetles, relating to their haemolymph containing repellent and toxic chemical compounds. They can deploy these chemical defences by “reflex bleeding” (adults and larvae) when disturbed and by coating laid eggs with their haemolymph. This defensive fluid can repel insect predators and competing insect species [1] but may also be toxic when ingested by other insects [2,3,4] and birds [5]. Alkaloids that are produced endogenously by the ladybirds comprise one component of these defensive secretions and are thought to account for this toxicity [6,7].

Historically, ladybirds have been used as a home analgesic remedy to soothe toothache [8]. Numerous other alkaloids from plants and animals have been used as therapeutic or pesticidal agents by targeting ion channels, including nAChRs [9,10], leading to the hypothesis that ladybird alkaloids could target nAChRs. This was confirmed when several azaphenalene alkaloids from multiple ladybird beetle species (Figure 1a) were found to inhibit mammalian nAChRs [11] and their enantioselective synthesis has since been described [12,13]. More recently, alkaloid extract from the Harlequin ladybird, *Harmonia axyridis*, containing 90% harmonine (Figure 1a), was shown to potently inhibit nAChRs from both insects and mammals [14].

The nAChRs are members of the Cys-loop family of ligand-gated ion channels [15] forming homo- or hetero-pentameric, transmembrane complexes of 5 highly homologous subunits, further sub-divided into α and non-α types. The α subunits (minimum of two) characteristically possess a pair of adjacent cysteine residues in loop C, one of the 6 loops (A–F) that make up the ACh binding site located at the interface between adjacent subunits [16]. Typically loops A, B and C are provided by the α subunit and loops D, E and F are donated by the non-α subunits. In vertebrates, 10 α, 4 β, 1 γ, 1 δ and 1 ε subunits are distributed throughout the nervous system [17] and skeletal muscle [18], where they exist in specific combinations relevant to their location and are predominantly associated with synaptic or neuromuscular transmission.

nAChRs are also found in insects where they are confined to the nervous system and play important roles in behaviour, as ACh is the primary afferent neurotransmitter in these organisms [19]. In insects, nAChRs have become important targets for insecticides [20,21,22]. These include Cartap [23], the commercial derivative of the natural marine annelid toxin, nereistoxin [24]; spinosyns [25]; and neonicotinoids, which bear structural similarities to the alkaloids nicotine and epibatidine [26]. Flupyradifurone, a butenolide [27], and Sulfoxaflor, a sulfoximine [28,29], agonists of insect nAChRs, were developed later, while triflumezopyrim is a recent addition to insecticides targeting insect nAChRs acting as an antagonist at the ACh binding site [30].

Investigation of natural products has proved to be a successful option for identifying novel insecticide leads [31] as well as providing therapeutic agents. As ladybird alkaloids have been shown to target both insect and mammalian nAChRs [11,14], there is clear value in investigating the activity of other ladybird alkaloids at nAChR, with subtype selectivity as a desirable outcome. Here, we have evaluated whether (−)-adaline, a structurally distinct alkaloid (Figure 1a) found in the haemolymph of *Adalia bipunctata* (Figure 1b) and *Adalia decempunctata*, has any agonistic or antagonistic activity at a range of nAChR subtypes, including human α1β1γδ natively expressed in TE671 cells; subtypes natively expressed in locust brain neurons; mammalian α7, α4β2 and α3β4, and a *Drosophila*/chicken hybrid receptor (Dα2/Gβ4) expressed in *Xenopus laevis* oocytes.

## 2. Results

### 2.1. Quantification of Adalia Alkaloid Extracts and Their Effects on Human Muscle and Locust Neuronal nAChRs

Extraction of alkaloids from adult *A. bipunctata* and *A. decempunctata* beetles resulted in an alkaloid extract containing (−)-adaline (Figure 1a) as a major component (Appendix A). The quantity of alkaloid extract obtained per beetle was relatively high for both *A. bipunctata* (0.79% of body weight) and *A. decempunctata* (0.69% of body weight) compared to that from several other ladybird species (Figure 2a).

The effect of *A. decempunctata* alkaloid extract was tested against human muscle-type nAChRs expressed in TE671 cells and insect neuronal-type nAChRs in locust neurons using whole-cell patch-clamp recording. No inward currents were detected upon application of the alkaloid extract alone to TE671 cells and locust neurons at a V_H_ of −75 mV (data not shown). However, when the alkaloids were co-applied with ACh (10 μM for TE671 cells or 100 μM for locust neurons), inhibition of the ACh response was observed in both TE671 cells and locust neurons (Figure 2b). Concentration-inhibition plots (Figure 2c) were created in order to obtain alkaloid extract IC_50_ values for inhibition of both the peak current amplitude and the current amplitude after 1 s (1-s current) (Table 1). Both were measured because some antagonists, notably Philanthotoxin-343, can have a much larger effect on the late current than the peak current [32], implying that inhibition was dependent on the activation of the receptor. The 1-s IC_50_ was 4.75-fold lower (*p* = 0.0019) than the peak IC_50_ for TE671 cells and 12.6-fold lower (*p* < 0.0001) for locust neurons. This implies that the inhibition was activation-dependent. It was also evident here that the inhibition of locust neuron nAChRs was greater than that of TE671 cells; the 1-s IC_50_ for locust neurons was 35.2-fold lower (*p* < 0.0001) than that for TE671 cells and the peak current IC_50_ was 13.2-fold lower (*p* < 0.0001) for locust neurons.

### 2.2. (−)-Adaline Inhibits Natively Expressed nAChRs in a Non-Competitive and Voltage-Dependent Manner

Application of (−)-adaline alone to whole-cell patch clamped TE671 cells and locust neurons resulted in no current responses at a V_H_ of −75 mV (data not shown). When co-applied with ACh (10 μM for TE671 cells or 100 μM for locust neurons), strong inhibition of the ACh response was observed (Figure 3). Concentration-inhibition plots for inhibition of peak current or 1-s current (Figure 3) revealed that the IC_50_ (Table 2) was much lower for 1-s current, indicating activation/use-dependence. Additionally, when repeated at a range of V_H_ (+50 mV to −120 mV for TE671 cells; −50 mV to −120 mV for locust neurons), the IC_50_ values were found to be voltage-dependent with lower values observed at more negative V_H_ (Table 2 and Figure 3). For example, the IC_50_s (1-s current) were 4.3-fold and 5.4-fold lower at −120 mV compared to −50 mV for TE671 cells and locust neurons, respectively, and at +50 mV for TE671 cells there was virtually no inhibition (Figure 3b,c). Together, this implied that inhibition was by an open channel blocking mechanism.

It was also apparent that the IC_50_ values for inhibition of locust neuronal nAChRs were significantly lower than those for TE671 cell nAChRs at all V_H_ (*p* < 0.0001 for all), indicating strong selectivity for this receptor type (Figure 4a; Table 2).

We further analysed the voltage-dependence by applying the Woodhull Equation (3) [33] to predict the depth of binding of (−)-adaline in the nAChR pore. Plots of IC_50_ (1-s) vs. V_H_ revealed a significantly sloping relationship for inhibition of both TE671 cell and locust neuronal nAChR (*p* = 0.0015 and *p* = 0.0022, respectively) (Figure 4b). The slopes of the relationships predicted that the fractional distance of the electric field traversed (δ) when (−)-adaline binds was 0.62 for TE671 cells and 0.74 for locust neurons.

Further mode of action studies were conducted solely on TE671 cells due to the low availability of (−)-adaline and experimental efficiency with these cells. ACh concentration-response plots in the presence and absence of 20 μM (−)-adaline showed that the ACh EC_50_ for peak and 1-s current was slightly increased, but in both cases this was not statistically significant (Figure 4c; Table 3). However, the maximum response to ACh was significantly reduced (Figure 4c) by 15% for peak current and 26% for 1-s current (*p* < 0.0001 for both) in the presence of 20 μM (−)-adaline. This supports a predominantly non-competitive mode of action for (−)-adaline inhibition of human muscle-type nAChRs in TE671 cells. Comparison of inhibition by (−)-adaline at each ACh concentration (Figure 4c) showed little dependence on ACh concentration; although, it was significantly greater at 10 μM compared to 300 μM ACh (*p* = 0.042) for peak current, and at 1 μM compared to 10 μM (*p* = 0.029) and 300 μM (*p* = 0.018) ACh for 1-s current. This suggests that there may also be a minor component of competitive antagonism.

### 2.3. Exploring Selectivity by Comparing Actions on Receptor Subtypes with Different Subunit Composition

The *Xenopus laevis* oocyte expression system and two electrode voltage-clamp technique were used to further investigate the selectivity between mammalian and insect nAChRs inhibition by (−)-adaline. Human α7, rat α4β2, rat α3β4 and the hybrid *Drosophila* α2/chicken β2 (Dα2/Gβ2) nAChRs were expressed in oocytes and (−)-adaline IC_50_s determined for inhibition of responses to ACh at V_H_ = −75 mV. All receptor types were inhibited by (−)-adaline (Figure 5). IC_50_ values obtained from concentration-inhibition plots (Figure 6a,b; Table 4) showed that inhibition was strongest for the rat α3β4 combination with the peak current IC_50_ being 13-fold, 48-fold and 113-fold lower (*p* < 0.0001 for all) than those for Dα2/Gβ2, α7 and α4β2, respectively (Table 4). For a more realistic comparison with the 1-s current measurement from the TE671 cell and locust neuron studies, the current was measured 15 s from response onset. This could only be achieved with the more slowly desensitising rat α4β2 and rat α3β4 nAChRs; α7 and Dα2/Gβ2 completely desensitise within this time period (Figure 5). The 15-s current IC_50_ values (Table 4) upheld the strong selectivity (139-fold; *p* < 0.0001) for inhibition of α3β4 over α4β2.

For each of the mammalian nAChR subunit combinations, the inhibition by 10 μM (−)-adaline was determined at several ACh concentrations (according to the sensitivity of each type). There were no significant differences in the level of inhibition across the ACh concentration ranges tested (Figure 7), supporting the previous finding that (−)-adaline is a non-competitive antagonist of nAChRs.

### 2.4. Invertebrate Bioassays

Contact assays with (−)-adaline caused species-specific mortality in both the susceptible and pyrethroid/organophosphate resistant tobacco whitefly strains and the mustard beetle (Table 5). The feeding assay showed that diamondback moth larvae were not substantially killed (only 3% mortality), however, there was strong anti-feeding activity with on average only 10% of the leaf disc consumed compared to 100% for untreated discs (Table 5).

## 3. Discussion

We have demonstrated that (−)-adaline, a major alkaloid component found in the haemolymph of 2-spot (*A. bipunctata*) and 10-spot (*A. decempunctata*) ladybirds, is a potent antagonist of various nAChRs tested here, particularly of insect types and the vertebrate ganglionic type, α3β4. It is speculated that these ladybirds use (−)-adaline as a defensive chemical, likely against invertebrate predators, and this may explain the selectivity shown towards the locust nAChRs studied here. It also implies that the nAChR is the natural target for the defensive chemical, a target also exploited by several classes of invertebrate pesticides [34,35]. We also extend the list of ladybird alkaloids that are known to target nAChRs, including the azaphenalenes from numerous species [11,12,13] and the diamine harmonine from the Harlequin ladybird, *H. axyridis* [14] (Figure 1). (−)-adaline is structurally diverse compared to these other ladybird alkaloids having a unique carbonyl group in its structure (Figure 1). This initially led us to believe that it may be an agonist of nAChRs but our experiments clearly demonstrate a lack of agonism and strong antagonism.

Inhibition of the ACh responses showed a pronounced time/activation dependence and this was particularly evident in those of TE671 cells and locust neurons where fast agonist application was possible. This was demonstrated by the greater inhibitory effect on the current 1 s after application (or 15 s after application for *Xenopus* oocyte experiments) and indicates that (−)-adaline inhibition is at least partly activation-dependent. This time-dependency may be because its binding site is only revealed in the open conformation of the nAChR, or it could be through acceleration of the desensitisation process [32]. Our further experiments showing that inhibition in TE671 cells and locust neurons was voltage-dependent suggests that the former explanation is most likely [32]. The fact that (−)-adaline was a more potent inhibitor at more negative V_H_ (Figure 3; Table 2) is consistent with its action as an open channel blocker. Further analysis of the voltage-dependence using the Woodhull equation (Figure 4b) [33] revealed δ values (fractional distance across the membrane electric field) of 0.62 to 0.74 implying that (−)-adaline binds beyond the equatorial leucine gate that is central in the nAChR pore, likely the serine and threonine residues that line the deeper narrow parts of the pore. In addition to this, at the positive V_H_ of +50 mV that was achievable in TE671 cells, there was negligible reduction of the ACh response (Figure 3b,c) indicating that the majority of the inhibition was via this open channel blocking mechanism. This is supported by our observations that (−)-adaline inhibition was dominantly non-competitive with the ACh EC_50_ not significantly increasing in the presence of (−)-adaline and its inhibition being largely independent of ACh concentration (Figure 4c and Figure 7).

The investigation of several azaphenalene alkaloids from other species of ladybird beetle show that they displaced binding to the *Torpedo* nAChR of tritiated piperidyl-*N*-(1(2-thienyl)cyclohexyl)-3,4-piperidine ([^3^H]-TCP), which binds deep within the channel pore, but did not displace binding of [^3^H]-cytisine, which binds to the ACh binding site [11]. The effects of the azaphenalene alkaloids, precoccinelline and coccinelline, were also examined on muscle-type nAChRs expressed in TE671 cells and precoccinelline on human α7 nAChRs expressed in *Xenopus* oocytes. Non-competitive inhibition was observed in each case [11]. This resembles our own observations on the binding of (−)-adaline to the pore region of nAChRs. Similarly, nereistoxin, an alkaloid from a marine annelid, and the insecticide derived from it, cartap, are also nAChR inhibitors that displace [^3^H]-TCP binding in honeybee (*Apis melifaria*) head membranes [23]. Philanthotoxin-343 (PhTX-343) derived from PhTX-433 from the Egyptian digger wasp *Philanthus triangulum*, and its numerous analogues also behave in a remarkably similar way to (−)-adaline [32,36,37,38], particularly with respect to the strong selectivity for α3β4 receptors [39]. This selectivity has been attributed to a valine to phenylalanine substitution uniquely found in the outer pore region of the vertebrate β4 subunit and this may contribute to strong (−)-adaline binding [39]. Interestingly, the same site is occupied by methionine in insect α-subunits and this may explain the observed selectivity for the locust neuron nAChRs [36].

Our finding that the alkaloid extract from *A. decempunctata* was similarly or slightly less effective as a nAChR inhibitor might imply that (−)-adaline is a major component of the extract. However, it could also indicate that the other known alkaloid, adalinine, is a nAChR inhibitor too but this remains to be determined. Furthermore, it is estimated that there are over 4200 species of ladybird beetle [40] but a relatively small number of these species have been investigated and alkaloids with a variety of different structures have been identified [41,42]. Therefore, investigation into alkaloids from other species could yield many more chemicals of interest.

## 4. Materials and Methods

### 4.1. Chemical Reagents and Nucleic Acids

ACh and all other common chemical reagents were obtained from Sigma-Aldrich unless otherwise noted. cDNA clones of rat neuronal nAChR subunits (α3, α4, β2, β4) were acquired from the laboratory of Professor Stephen Heinemann at the Salk Institute for Biological Studies. The cDNA clones for the human α7, the *Drosophila* Dα2 and the chick β2 nAChR subunits were a gift from Professor David Sattelle, University College London. Plasmids were linearised and cRNA transcribed using the mMessage mMachine transcription kit (Ambion, Austin, TX, USA).

### 4.2. Extraction of (−)-Adaline

Adult two spot ladybirds, *Adalia bipunctata,* were either collected from the grounds of the University of Nottingham (University Park, Nottingham, UK), or were purchased from Syngenta Bioline (Little Clacton, UK). All ladybirds were stored at −20 °C until required for extraction. Adults (*n* = 628, 4.62 g) were frozen using liquid nitrogen, ground and extracted using 250 mL methanol (Honeywell, Offenbach am Main, Germany) in an Erlenmeyer flask at ambient temperature for 24 h. The contents were extracted for a further 24 h, and the collected extracts combined then evaporated in vacuo to yield a residue. The residue was subjected to acid-base extraction (1M HCl (50 mL), wash with diethyl ether (50 mL), pH of the collected aqueous layer adjusted to 10–12 using 2M NaOH, then extracted with dichloromethane (Fisher Scientific, Loughborough, UK, 2 × 50 mL)). The organic layers were combined, washed with saturated NaCl solution (10 mL), dried using anhydrous magnesium sulphate (MgSO_4_), filtered and evaporated in vacuo to yield a crude alkaloid extract (36.7 mg), which was stored in a glass ampoule sealed under nitrogen at 4 °C until required for liquid chromatography. An aliquot of the crude extract was subjected to small-scale liquid chromatography over neutral alumina as described in Lognay et al., 1996 [43]. Chloroform: hexane (1:1) was used as the eluant to obtain fractions, which were shown by Thin Layer Chromatography (TLC; aluminium backed silica plates) and Dragendorff’s reagent to contain alkaloids [44]. Fractions shown by GC-MS analysis to contain (−)-adaline (see Appendix A for GC-MS analysis method and data) were collected and combined to yield a colourless oil, which was aliquoted and stored in glass ampoules sealed under nitrogen at 4 °C until required for laboratory assays.

### 4.3. TE671 Cell and Locust Neuron Culture

Human TE671 cells are known to express the embryonic form of muscle-type nAChRs (containing (α1)_2_β1δγ subunits) [45]. Cells were maintained in growth medium containing Dulbecco’s modified Eagle’s medium (DMEM; 4.5 g/L glucose) supplemented with 10% fetal calf serum, 2 mM glutamine, 10 IU/mL penicillin and 10 μg/mL streptomycin; and incubated at 36.5 °C in a 5% CO_2_ atmosphere. Cells were grown in 25 cm^2^ flasks and divided 1:10 when they were approximately 75% confluent. For whole-cell patch-clamp electrophysiological recordings, dividing cells were plated onto pieces of glass coverslip (5 × 20 mm) in 35-mm Petri dishes containing 2 mL growth medium.

Desert locusts (*Schistocerca gregaria*) were purchased from Livefoods UK Ltd., Axbridge, UK and kept in locust breeding cages maintained at an ambient temperature of 26–28 °C and a 12:12 h light cycle. Locusts were selected at the 6th instar, cold anaesthetised at 4 °C for 10 min, dipped in 70% ethanol and decapitated. Locust heads were transferred to cooled Ca^2+^/Mg^2+^ free Rinaldini’s saline (135 mM NaCl, 25 mM KCl, 0.4 mM NaHCO_3_, 0.5 mM glucose, 5 mM HEPES, pH 7.2 with NaOH). Brains were removed and placed in cooled Ca^2+^/Mg^2+^ free Rinaldini’s saline. The mushroom bodies were dissected and placed in 200 μL of Rinaldini’s saline containing 2 mg/mL collagenase (Type 1A) and 0.5 mg/mL dispase (Boehringer Mannheim UK Ltd., Welwyn Garden City, UK). After 15 min incubation at 36.5 °C, the tube was centrifuged at 800× *g* for 1 min at room temperature. The supernatant was removed and replaced with 200 μL locust culture medium (5:4 DMEM [supplemented with 10% FCS, 2 mM Glutamine]:Schneider’s insect medium, with 10 IU/mL penicillin and 20 μg/mL streptomycin). The mushroom bodies were then gently triturated through a 200 μL pipette tip before being distributed over heat-sterilised glass coverslips (5 × 20 mm) coated in 0.01% poly-l-lysine (Sigma-Aldrich, Gillingham, UK) that had been placed into 35 mL Petri dishes containing 2 mL locust culture medium. Dishes were incubated at 36.5 °C in a 5% CO_2_ atmosphere and used within 24 h.

### 4.4. Whole-Cell Patch-Clamp Electrophysiology

Whole-cell patch-clamp electrophysiology was carried out on TE671 cells and locust neurons using an Axopatch 200A (Molecular Devices, San Jose, CA, USA) patch-clamp amplifier and recorded to the disk of a PC using an NI PCI-6221/BNC-2110 data acquisition system (National Instruments, Austin, TX, USA) controlled by WinWCP software (Dr. John Dempster, Institute of Pharmacy & Biomedical Sciences, University of Strathclyde, Glasgow, UK). Patch-pipettes were formed using borosilicate glass capillaries (1B150F-4, World Precision Instruments, Hitchin, UK) using a programmable micropipette puller (P-97, Sutter Instruments Co., Novato, CA, USA) giving resistances of 5–7 MΩ when filled with pipette solution containing 140 mM CsCl, 1 mM MgCl_2_, 11 mM EGTA and 5 mM HEPES, pH 7.2. Solutions were perfused using a DAD-12 Superfusion system (ALA Scientific Instruments, Farmingdale, NY, USA) fitted with a 100 μm polyamide coated quartz output tube with a solution exchange time of 30–50 ms and controlled by WinWCP. The perfusion system was pressurised with compressed nitrogen and solutions applied at 200 mm/Hg. TE671 cells were placed in a perfusion chamber and constantly perfused at a flow rate of 5 mL/min with mammalian saline solution (135 mM NaCl, 5.4 mM KCl, 1 mM CaCl_2_, 1 mM MgCl_2_, 5 mM HEPES, 10 mM d-glucose, pH 7.4 with NaOH). Locust neuronal cultures were perfused with locust saline (180 mM NaCl, 10 mM KCl, 2 mM CaCl_2_, 10 mM HEPES, pH 7.2 with NaOH).

### 4.5. Xenopus Laevis Oocyte Preparation and cRNA Injection

*Xenopus laevis* oocytes were acquired from the European Xenopus Resource Centre (University of Portsmouth, Portsmouth, UK). On arrival, ovary tissue was treated with 0.5 mg/mL collagenase (Type 1A) in Ca^2+^-free modified Barth’s saline (MBS) (96 mM NaCl, 2 mM KCl, 5 mM HEPES, 2.5 mM pyruvic acid & 0.5 mM theophylline, pH 7.5) for 1 h at 18 °C to release individual cells and remove the follicular tissue surrounding the oocytes. After washing with Ca^2+^-free MBS the isolated oocytes were incubated at 18 °C in MBS (96 mM NaCl, 2 mM KCl, 1.8 mM CaCl_2_, 5 mM HEPES, 2.5 mM pyruvic acid & 0.5 mM theophylline, 0.05 mg/mL gentamicin, pH 7.5). Healthy stage IV-V oocytes were selected and 50 nL cRNA was injected using a Nanoliter injector (World Precision Instruments, UK). Human α7 was injected at a concentration of 100 ng/μL, rat α3/β4 and rat α4/β2 were injected in a 1:1 ratio at a concentration of 200 ng/μL each and *Drosophila* α2/chick β2 was injected in a 1:1 ratio at a concentration of 1 μg/μL each. Oocytes were incubated for 3–4 days at 18 °C prior to electrophysiological recordings.

### 4.6. Two-Electrode, Voltage-Clamp Electrophysiology

Whole-cell current recordings were obtained from nAChR-expressing oocytes by two-electrode voltage clamp using an Axoclamp 2A voltage clamp amplifier (Molecular Devices, USA). An oocyte was transferred to the perfusion chamber and perfused (~5 mL/min) with standard oocyte saline (SOS) (100 mM NaCl, 2 mM KCl, 1.8 mM CaCl_2_, 1 mM MgCl_2_, 5 mM HEPES, pH 7.6). Microelectrodes were pulled from borosilicate glass capillaries (GC150TF-4, Harvard Apparatus, Cambridge, UK) using a programmable micropipette puller (P-97, Sutter Instruments Co., USA), having resistances between 0.5 and 2.5 MΩ when filled with 3 M KCl. The oocyte was voltage-clamped at a holding potential (V_H_) of −75 mV. ACh was consistently used as the agonist, and it was applied without or together with (−)-adaline via a MPS-2 multi-channel gravity fed perfusion system (World Precision Instruments, UK). Atropine (0.5 μM) was added to the SOS to prevent any endogenous muscarinic ACh receptor response [46]. Output currents were transferred using an NI PCI-6221/BNC-2110 A/D converter (National Instruments, USA) to a PC and WinEDR software (Dr John Dempster, Institute of Pharmacy & Biomedical Sciences, University of Strathclyde, UK) was used for recording and analysis.

### 4.7. Invertebrate Bioassays

Strains of invertebrate pests, *Musca domestica* L. (housefly); *Phaedon cochleariae* Fab. (mustard beetle); *Myzus persicae* Sulzer (peach-potato aphid; susceptible strain USIL); *Bemisia tabaci* Genn. (tobacco whitefly, susceptible strain SUD-S; pyrethroid and organophosphate insecticide resistant strain ISR-R); *Tetranychus urticae* (red spider mite, susceptible strain UK-S) and *Plutella xylostella* L. (diamondback moth) were from established laboratory cultures at Rothamsted Research. General procedures and contact bioassay protocols used in this study involving topical application/microimmersion have been described in full elsewhere [47]. In the feeding bioassay for *P. xylostella,* fresh leaf discs from Chinese cabbage were coated with 200 μL of (−)-adaline solution and air dried before transferring 3rd instar larvae to them. Mortality and feeding damage (expressed as% of leaf area consumed) was assessed after 48 h. In all cases, observed mortalities were corrected for control mortality using Abbott’s formula. Test solutions of (−)-adaline were prepared in acetone. All dose–response assays encompassed at least four test concentrations with a minumum of two replicates of 15 individuals per concentration. Data were subjected to probit analysis to obtain LC_50_ estimates.

### 4.8. Data Analysis

WinWCP software was used to measure the peak current and current 1 s after onset of response for patch clamp recordings. WinEDR software was used to measure the peak current and current 15 s after onset of response (15-s current) for two-electrode voltage clamp recordings. Data were normalised as% of an ACh control response or% of maximal ACh response. Each plotted data point is the mean ±SEM of recordings from 5–18 cells or 5–8 oocytes. GraphPad Prism 9 was used for all data analysis, graph plotting, curve fitting and statistical tests. Concentration-inhibition and concentration-response curves were used to calculate IC_50_s for (−)-adaline or EC_50_s for ACh, respectively, using the following equations:(1)$$\%\;control\;response=100\left(1+10^{\left(LogIC_{50}-X\right)\times Hillslope}\right)$$
or
(2)$$\%\;max\;ACh\;response=max\left(1+10^{\left(LogEC_{50}-X\right)\times Hillslope}\right)$$
where *X* is the Log concentration of ACh or (−)-adaline. The voltage-dependence of inhibition by (−)-adaline was further analysed using the Woodhull equation [33]:(3)$$IC_{50}\left(V_{H}\right)=IC_{50}\left(0\right)e^{\left(\frac{z\delta VF}{RT}\right)}$$
where *δ* is the fraction of the membrane electric field sensed by the blocker as it binds, *R* is the gas constant, *T* is the absolute temperature, *F* is Faraday’s number and *z* is the valence of the blocker (=1 for adaline).

## 5. Conclusions

In conclusion, we demonstrate that (−)-adaline, a natural alkaloid with novel structural properties, is an open channel blocker of various nAChRs. Its apparent selectivity for insect nAChRs may be useful as a lead for insecticide development, whilst its strong selectivity for α3β4 receptors amongst vertebrate nAChRs may give it therapeutic relevance, for example, this nAChR subtype has been proposed as a target for smoking cessation [48,49] or reduced opioid self-administration [50].

## Figures and Tables

**Figure 1 molecules-27-07074-f001:**
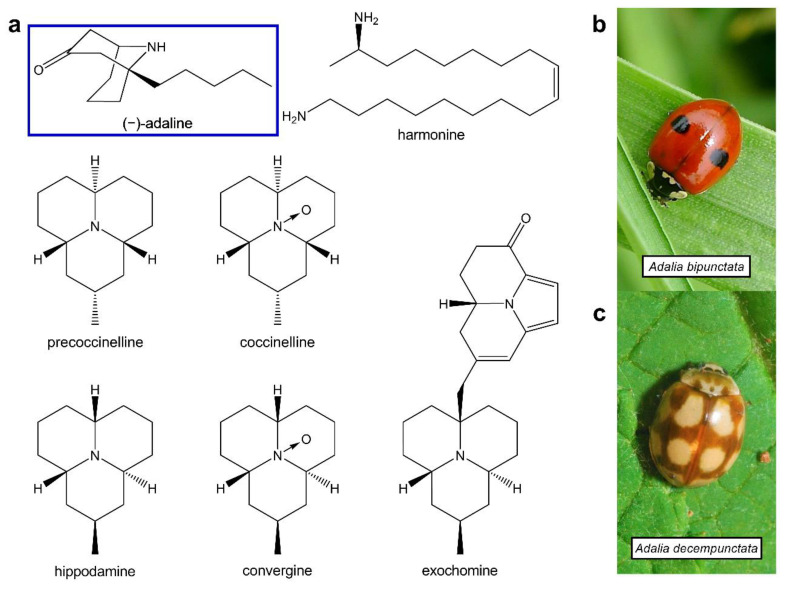
Ladybird beetle alkaloids with known nAChR activity. (**a**) The structure of (−)-adaline from *A. bipunctata* (**b**); Gail Hampshire, https://www.flickr.com/photos/gails_pictures/8357339437 (accessed on 17 August 2022) and *A. decempunctata* (**c**) alongside alkaloids from other ladybird species that have inhibitory action at nAChRs.

**Figure 2 molecules-27-07074-f002:**
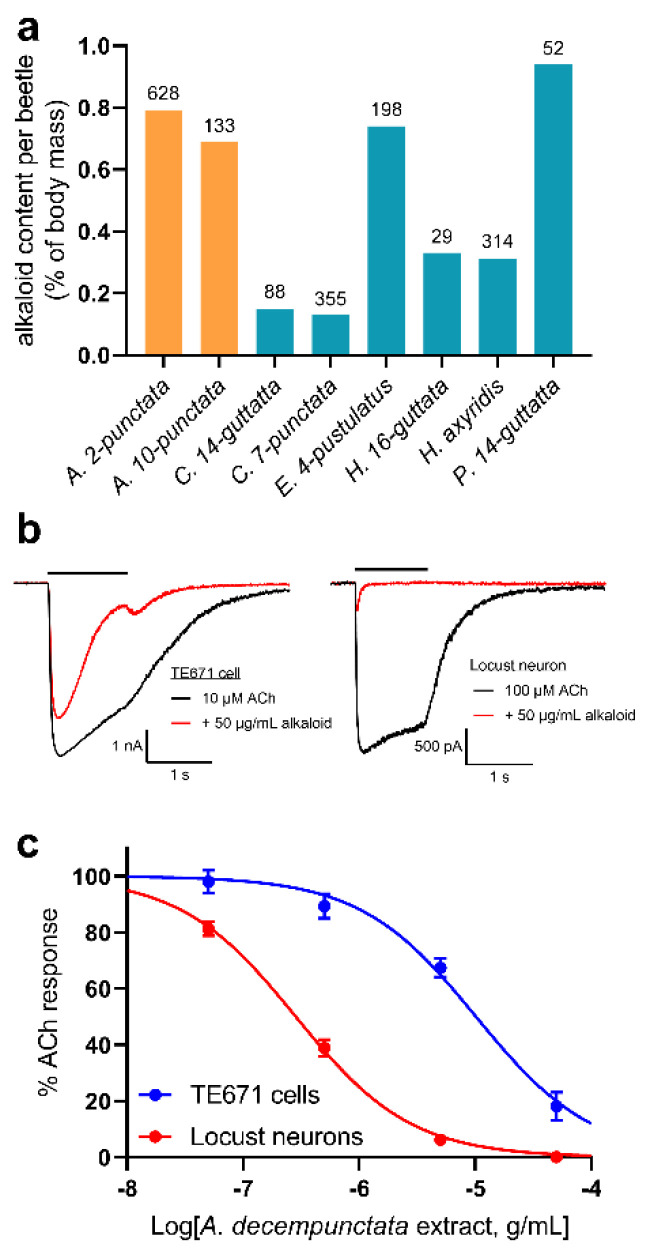
Inhibitory action of *A. decempunctata* alkaloid extract. (**a**) Alkaloid content per *A. bipunctata* or *A. decempunctata* beetle (orange bars) compared to that of several other ladybird species (turquoise bars) using similar extraction procedures. Amounts are total alkaloid extract/number of beetles used in the extraction (numbers above bars). (**b**) Representative traces showing the inhibition of ACh-induced whole-cell currents in both TE671 cells (**left**) and locust neurons (**right**) by co-applied 50 μg/mL *A. decempunctata* alkaloid extract at V_H_ = −75 mV. (**c**) Concentration-inhibition curves for *A. decempunctata* alkaloid extract inhibition of nAChR currents at 1 s in TE671 cells and locust neurons (V_H_ = −75 mV), showing selectivity for locust over TE671 nAChR. Points are mean% control response to 10 μM (TE671 cells) or 100 μM (locust neurons) ACh and error bars are SEM (*n* = 10–15). Curves are fits of Equation (1) and IC_50_s derived from this are given in Table 1.

**Figure 3 molecules-27-07074-f003:**
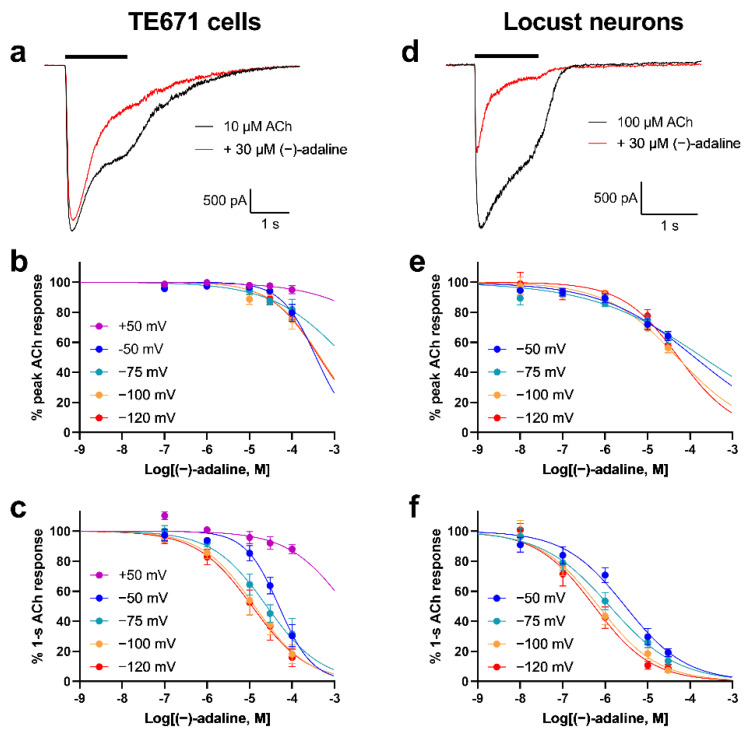
(−)-adaline causes time and voltage dependent inhibition of nAChRs in TE671 cells and locust neurons. Representative traces showing the inhibition of ACh-induced whole-cell currents in both TE671 cells (**a**) and locust neurons (**d**) by co-applied 30 μM (−)-adaline at V_H_ = −75 mV. Concentration-inhibition curves for (−)-adaline inhibition of nAChR peak (**b,e**) and 1-s (**c**,**f**) currents in TE671 cells (**b**,**c**) and locust neurons (**e**,**f**) at a range of V_H_. Points are mean% control response to 10 μM (TE671 cells) or 100 μM (locust neurons) ACh and error bars are SEM (*n* = 5–18). Curves are fits of Equation (1) and IC_50_s derived from this are given in Table 2.

**Figure 4 molecules-27-07074-f004:**
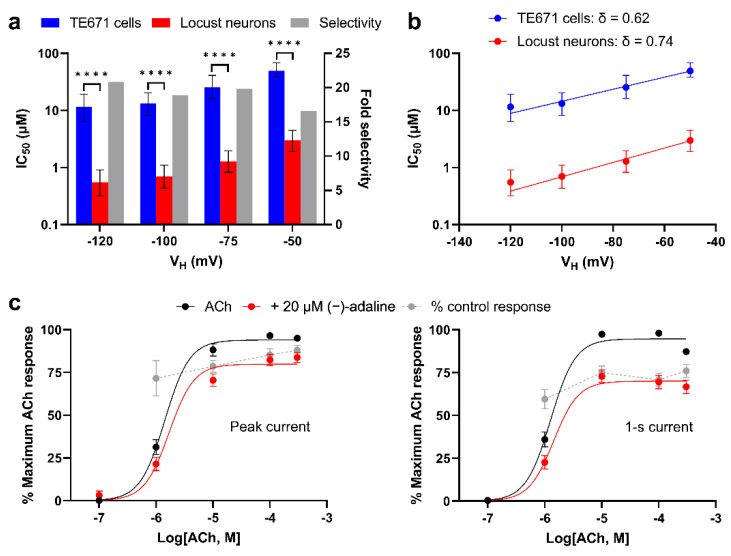
Inhibition by (−)-adaline is voltage-dependent, non-competitive and selective for locust nAChR. (**a**) Comparison of 1-s current IC_50_s for (−)-adaline inhibition of nAChRs between TE671 cells and locust neurons at each tested V_H_. Blue and red bars indicate IC_50_ and error bars are the 95% CI (left *Y*-axis). Grey bars show selectivity (right *Y*-axis) for locust neuronal nAChR over human muscle nAChR; in all cases *p* < 0.0001 (****). (**b**) IC_50_s (with 95% CI) for 1-s data were plotted against V_H_ and fitted with Equation (3). The slopes of the lines (δ) were both significantly greater than 0 (*p* = 0.0015 and 0.0022) confirming voltage-dependent inhibition by (−)-adaline of both TE671 cell and locust neuronal nAChRs. The δ-values indicate that (−)-adaline can bind deep in the pore beyond the equatorial leucine gate. (**c**) Concentration-response plots showing the peak (left) and 1-s (right) current response of TE671 cells to ACh alone (black) and to ACh co-applied with 20 µM (−)-adaline (red) at V_H_ = −75 mV. Points are the mean% of the maximum response to ACh alone and error bars are SEM (*n* = 16). There was a significant (both *p* < 0.0001) reduction in maximum response to ACh. ACh EC_50_s are given in Table 3 but did not significantly increase in the presence of (−)-adaline. Furthermore, plotted in grey are the mean% of control ACh response at each ACh concentration (±SEM, *n* = 16) to evaluate any dependence of inhibition on ACh concentration.

**Figure 5 molecules-27-07074-f005:**
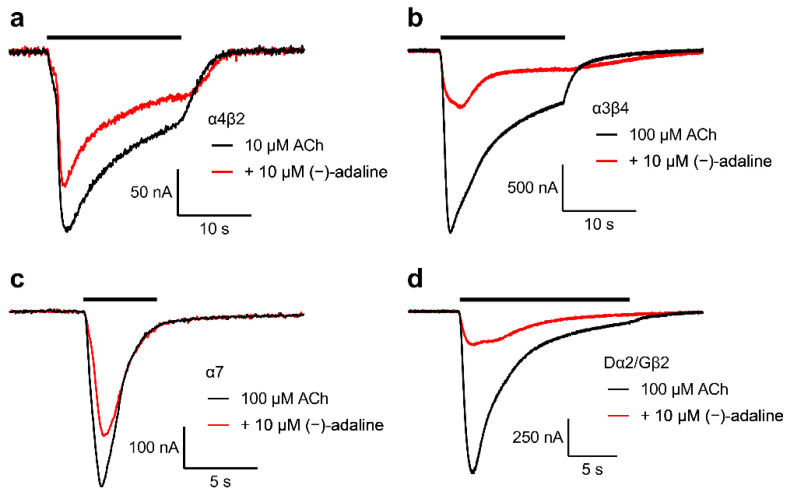
(−)-adaline inhibits the ACh response of 3 well characterised mammalian neuronal nAChRs and a hybrid insect/chicken nAChR expressed in *Xenopus* oocytes. Representative traces showing the inhibition by 10 μM (−)-adaline of ACh-induced currents in the rat α4β2 (**a**), rat α3β4 (**b**), human α7 (**c**) and hybrid Dα2/cGβ2 (**d**) nAChRs expressed in *Xenopus* oocytes at V_H_ = −75 mV.

**Figure 6 molecules-27-07074-f006:**
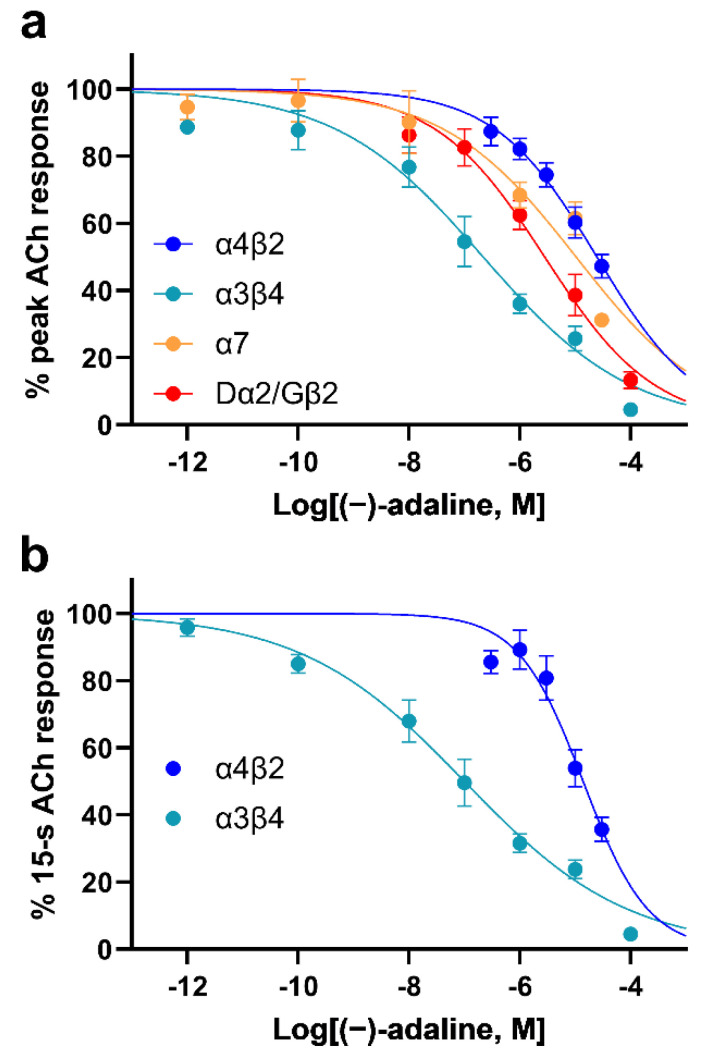
(−)-adaline shows selectivity for the rat α3β4 nAChR. Concentration-inhibition curves showing the actions of (−)-adaline on (**a**) the peak ACh-induced current for the hybrid rat α4β2, rat α3β4, human α7 and Dα2/Gβ2; and (**b**) on 15-s current responses for rat α4β2 and rat α3β4 expressed in *Xenopus* oocytes. Points are mean% control response to 10 μM (α4β2) or 100 μM (α3β4, α7, Dα2/Gβ2) ACh and error bars are SEM (*n* = 4–8). Curves are fits of Equation (1) and IC_50_s derived from this are given in Table 3.

**Figure 7 molecules-27-07074-f007:**
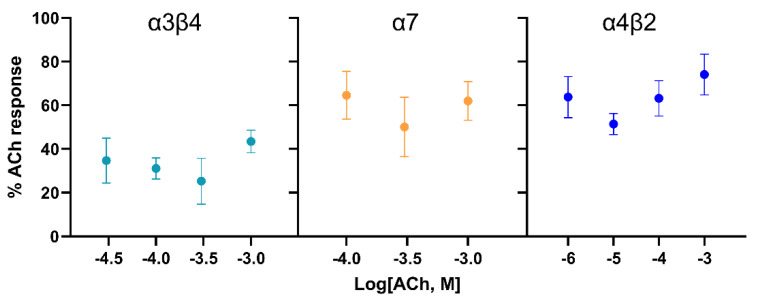
(−)-adaline inhibition of rat α3β4, human α7 and rat α4β2 nAChRs is non-competitive. Inhibition by (−)-adaline was assessed at several ACh concentration for each receptor subtype and was plotted as mean% of control ACh response against ACh concentration; error bars are SEM (*n* = 5–11).

**Table 1 molecules-27-07074-t001:** IC_50_ values for inhibition by *A. decempunctata* alkaloid extract of the peak and 1-s current of human muscle-type nAChRs expressed in TE671 cells and locust neuronal nAChRs.

Cell/nAChR Type	IC_50_, µg/mL (95% CI)		Peak/1-s Ratio
	Peak Current	1-s Current	
TE671 (human muscle-type)	48.4 (13.9–169)	10.2 (7.18–14.5)	4.75
Locust (insect neuronal-type)	3.66 (2.38–5.63)	0.29 (0.24–0.33)	12.6

**Table 2 molecules-27-07074-t002:** IC_50_ values for inhibition by (−)-adaline of the peak and 1-s current of human muscle-type nAChRs expressed in TE671 cells and locust neuronal nAChRs at different V_H_.

Cell/nAChR Type	V_H_ (mV)	IC_50_, µM (95% CI)		Selectivity
		Peak Current	1-s Current	
TE671 (human muscle-type)	+50	>>100	>>100	-
	−50	>100	49.2 (38.1–68.0)	
	−75	>100	25.4 (16.2–41.1)	
	−100	>100	13.2 (8.17–20.3)	
	−120	>100	11.5 (6.38–19.2)	
Locust (insect neuronal-type)	−50	>100	2.97 (1.92–4.51)	17
	−75	>100	1.28 (0.83–1.97)	20
	−100	50.3 (33.3–94.8)	0.70 (0.44–1.10)	19
	−120	53.4 (32.9–146)	0.55 (0.32–0.91)	21

**Table 3 molecules-27-07074-t003:** ACh EC_50_ values for the peak and 1-s current of human muscle-type nAChRs expressed in TE671 cells in the absence and presence of 20 μM (−)-adaline.

Current	EC_50_, µM (95% CI)		*p*-Value
	ACh Alone	+20 μM (−)-Adaline	
Peak	1.43 (1.26–1.64)	1.70 (1.37–2.26)	0.171
1-s	1.27 (1.14–1.42)	1.43 (1.15–1.83)	0.308

**Table 4 molecules-27-07074-t004:** IC_50_ values for inhibition by (−)-adaline of human α7, rat α4β2, rat α3β4 and the hybrid *Drosophila* α2/chicken β2 (Dα2/Gβ2) nAChRs expressed in *Xenopus* oocytes. V_H_ = −75 mV. N/A = no or negligible current after 15 s.

Receptor	IC_50_, μM (95% CI)	
	Peak Current	15-s Current
α4β2	24.8 (16.2–46.2)	14.0 (9.76–22.0)
α3β4	0.22 (0.10–0.47)	0.10 (0.047–0.21)
α7	10.4 (4.61–29.9)	N/A
Dα2/Gβ2	2.84 (1.60–5.03)	N/A

**Table 5 molecules-27-07074-t005:** Mortality of several pest invertebrate species following treatment with (−)-adaline.

Species	Activity
House fly	0% ^a^
Mustard beetle	85% ^a^
Tobacco whitefly (SUD-S)	12 ppm ^b^
Tobacco whitefly (ISR-R)	26 ppm ^b^
Green peach aphid	1% ^c^
Red spider mite	0% ^c^
Diamondback moth	3% ^d^
Diamondback moth	90% ^e^

^a^ % mortality for 2 μg (−)-adaline topical application; ^b^ LC_50_ for (−)-adaline; ^c^ % mortality for 1000 ppm (−)-adaline immersion; ^d^ % mortality by leaf disc treatment with 200 μL 1000 ppm (−)-adaline; ^e^ % uneaten of leaf disc treated with 200 μL 1000 ppm (−)-adaline.

## Data Availability

All data and supporting data are reported in this manuscript and Appendix A.

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
