# Peer review of "(−)-Adaline from the Adalia Genus of Ladybirds Is a Potent Antagonist of Insect and Specific Mammalian Nicotinic Acetylcholine Receptors"

_molecules, 2022, doi:10.3390/molecules27207074_

Round 1

Reviewer 1 Report

Congratulations to the authors on their manuscript entitled “(−)-Adaline from the Adalia genus of ladybirds is a potent antagonist of insect and specific mammalian nicotinic acetylcholine receptors”. In this study, several alkaloids were identified from ladybirds with nAChR activity, and particularly, (−)-adaline as a lead with potent antagonistic activity was carefully studied.
Overall, this is an interesting study, however, few concerns should be considered. In Fig 2C, it’s not enough to perform a nonlinear fitting to calculate IC50 with only 4 spots, more spots are required. Fig 6B has the same problem, why α3β4 has 7 spots while α4β2 has 5 spots. In SI, the gas chromatogram is not clear, please redraw the figure based on the original data.

Author Response

Response:

“In Fig 2C, it’s not enough to perform a nonlinear fitting to calculate IC50 with only 4 spots, more spots are required.” – We agree that ideally more points would be useful for a more accurate determination of IC50, however, the confidence interval given takes into account the degrees of freedom (includes number of points) as well as the cell to cell variability and the goodness of fit. Importantly, all of the points are on the sloping part of the relationship and furthermore the data are normalised such that we know the upper plateau is 100% and lower plateau is 0%, effectively adding two further points. So we are confident that the IC50s estimated are a good reflection of the data, despite the number of points, and the confidence intervals are clearly stated (in Table 1). It would be impossible to repeat the experiments with more points within the available timeframe for revision, especially given the scarcity of A. decempunctata and thus the alkaloid extract.

“Fig 6B has the same problem, why α3β4 has 7 spots while α4β2 has 5 spots.” – We are happy that 7 and 5 points are sufficient for good estimation of the IC50 for the same reasons as given above. They differ as activity against the two receptor subtypes takes place over different concentration ranges (wider for α3β4).

“In SI, the gas chromatogram is not clear, please redraw the figure based on the original data.” – We have provided a better reproduction of the gas chromatogram and added the coupled MS.

Reviewer 2 Report

The authors explored the actions of (−)-adaline, found in the 2-spot (Adalia bipunctata) and 10-spot (Adalia decempunctata) ladybirds, on both mammalian (α1β1γ, α7, α4β2, α3β4) and insect nAChRs using patch-clamp of TE671 cells and locust brain neurons natively expressing nAChRs, as well as two-electrode voltage clamp of Xenopus laevis oocytes recombinantly expressing nAChRs. All nAChR subtypes were antagonised by (−)-adaline in a time-dependent, voltage-dependent and non-competitive manner with the lowest IC50s at rat α3β4 (0.10 μM) and locust neuron (1.28 μM) nAChRs, at a holding potential of −75 mV. The data imply that (−)-adaline acts as an open channel blocker of nAChRs.

Strengths:
This study is novel and interesting, provides more insights for the mechanociception. The experimental design is overall rigorous, and the statistical analysis is good and test methods were chosen correctly. 

Weakness:

In figure 2A, there were no error bar in each column, does it mean n = 1? Could the authors explain it?

Author Response

Response:

“In figure 2A, there were no error bar in each column, does it mean n = 1? Could the authors explain it?” – The values represent the mean alkaloid content per beetle as % of body weight for a large number of individuals in each case (A. bipunctata n = 628, A. decempunctata n = 133, C.quatuordecimguttata n = 88, C. septempunctata n = 355, E. quadripustulatus n = 198, H. sedecimguttata n = 29, H. axyridis n = 314 and P. quatuordecimpunctata n = 52). However, it was not plausible to perform the alkaloid extraction from each individual to calculate the mean and SD or SEM. Instead, it was necessary to combine all of the individuals in a single extraction followed by dividing the total amount of alkaloid extracted by the number of individuals used in the extraction. Thus, even though n was >>1 beetles, it was not possible to get SD or SEM, and hence no error bars. We have altered the Y-axis title, added the n numbers above the bars and added a note to the figure legend to clarify.